# Using fractal model and factor analysis for FACA modeling and its application in deep mineral prediction

Feilong Qin[1,2], Hongjin Zhu[1]*, Yu Feng[1]*, ShiCheng Yu[1]

**1** School of Big Data and Artificial Intelligence, Chengdu Technological University, Chengdu, China,
**2** College of mathematics and Science, University of Electronic Science and Technology of China, Chengdu, China

\* zhjin2@cdtu.edu.cn (HZ); fengyu@cdtu.edu.cn (YF)

## Abstract

### Objective

This paper designs a FACA model for deep mineral prediction in actual mining areas.

### Method

The spatial distribution of geochemical anomalies was consistent with the concentration--area (C-A), this paper established a C–A model for geochemical anomaly extraction. Based on mineral resources formed by multiple element combinations, factor analysis (FA) was used to obtain the different combinations and comprehensive information of elements. On this basis, a new FACA model was designed for mineral prediction using the FA and C–A.

### Results

The proposed FACA model was applied to mineral prediction in the Jiguanzui copper-gold mining area in China. The elements in the study area were divided into four combinations. The thresholds of single element and element combination anomalies were identified. Through diagnostic testing, the abnormal distributions of geochemical elements were consistent with their theoretical distributions, and the comprehensive abnormal distribution area of elements was consistent with the distribution of the actual ore bodies, demonstrating that the designed FACA algorithm of this paper was reasonable.

### Conclusions

A new blind ore body in the study area is predicted by using FACA model, positioned at a depth ranging from approximately 1120m to 1150m below ground, between drill holes ZK02618 and KZK23. These findings hold significant implications for mineral exploration efforts.

**Data availability statement:** All relevant data are within the manuscript and its Supporting information files. The complete sample data of geochemical element contents can be seen at https://github.com/qflong1/Studying-data.git (Table 1). The complete standardized data (Table 3) can be seen at https://github.com/qflong1/Studying-data.git (Table 3). The complete factor scores data (Table 8) can be seen at https://github.com/qflong1/Studying-data.git (Table 8).

**Funding:** The study data was supported by the Opening Fund of Geomathematics Key Laboratory of Sichuan Province (Grant no. scsxdz2022-06), Program of Science & Technology Department of Sichuan Province (Grant no. 2026NSFSC0239) and the Program of Sichuan Mineral Resources Research Center (Grant no. SCKCZY2023-ZC006).

**Competing interests:** The authors have declared that no competing interests exist.

## Introduction

Mineral resources play an important role in society's development and are formed by the enrichment of geochemical elements in the earth [1,2]. When the distributions of geochemical element contents are deviated from the surrounding element contents, the geochemical anomalies are formed [3]. Therefore, we need to select a reasonable method for extracting geochemical anomalies. In recent years, scholars have conducted extensive research on the extraction of geochemical anomalies, such as cumulative frequency method, mean square deviation method, trend analysis method, multiple regression method, machine learning method, deep learning method, evidence weighting method, etc.[4,5], which have been widely used to identify geochemical anomalies. However, the cumulative frequency method, mean square deviation method, trend analysis method, and multiple regression method require the sample data to follow a normal distribution, the data of geochemical element contents does not follow a normal distribution because the geochemical elements are affected by various geological environments, so these traditional statistical methods for identifying geochemical anomaly is not effective [1,6]. The evidence-weighting method relies on expert experience to estimate the critical value of mineralization probability, lacks theoretical support, and may consequently result in the identification of pseudo anomalies unrelated to the ore body [7]. The effectiveness of geochemical data processing by using machine learning and deep learning methods is not ideal, because these methods require a large amount of data for training, while geochemical data is obtained through geological sampling and has a small amount of data [8,9]. At present, each method has required some conditions to identify geochemical anomalies, we thus need to find a reasonable method for identifying geochemical anomalies.

The distribution of geochemical element content data does not follow a normal distribution and has nonlinear characteristics, so traditional statistical methods are unable to process these data. The C–A model was proposed for processing nonlinear data by Chen et al. [10], a famous Academician of CAS. As the C–A model not only considers the frequency characteristics but also considers the spatial distribution characteristics of sample data [11], it has been widely used in many fields for nonlinear data processing [12,13]. For example, the fractal dimension has emerged as a crucial parameter for quantifying the complexity of pore structures [14] and revealing the intrinsic connections between micro-fractures, mesoscale fractures, and large-scale faults [15]. The C-A model has been effectively applied to the extraction of remote sensing alteration information [16], particularly in the identification of geochemical anomalies [17,18]. Drawing inspiration from the successful application of the C–A model, we have designed a geochemical anomaly extraction model with the C–A. This model is tailored for identifying anomalies in geochemical element contents within actual mining areas. By leveraging the spatial distribution of geochemical anomalies, obtained through the C–A model, in conjunction with geological conditions, we can predict the location of potential new ore bodies. Although C–A model has excelled in identifying anomalies of a single element, it lacks ability to process mineral resource data formed by the combined effects of multiple geochemical elements [19].

In fact, minerals are formed through the combination of a multitude of geochemical elements. Therefore, relying solely on the anomalous distribution of a single element for mineral prediction has inherent limitations. It is more sensible to utilize the abnormal distributions of comprehensive multi-element information to predict the location of ore bodies. Through our research, we discovered that FA [20]can not only extract comprehensive information about elements but also classify them. In recent years, FA, as a statistical method, has been extensively applied in various fields [21,22]. When FA is used to process geochemical data, the data are categorized into distinct combination categories, which correspond to different types of altered minerals [23]. FA has advantages in dimensionality reduction of geological data and identification of geological element associations, but the spatial distribution characteristics of geochemical anomalies are often neglected when using FA model alone for mineral prediction, which to some extent limits its prediction accuracy.

Given this, we designed a novel FACA model that integrates FA and C-A. The model fully leveraging the advantages of the C-A model in processing nonlinear data and extracting geochemical element anomalies, as well as the strengths of FA in geochemical element combination identification and data dimensionality reduction. Compared with previous studies, this study not only expands the application scope of the C-A model from single-element anomaly identification to multi-element comprehensive information extraction, but also enhances the utilization efficiency of element combination information in mineral prediction by introducing the FA model. Furthermore, the FACA model also considers the spatial distribution characteristics of geochemical anomalies, significantly improving the accuracy and reliability of predictions by combining geological conditions with mineral prediction. Therefore, this study not only innovates in methodology by proposing a new mineral prediction model by integrating fractal models with factor analysis, but also demonstrates its effectiveness in practical applications, providing new ideas and tools for deep mineral prediction, which is crucial for in-depth mineral prediction.

## Methods

### C–A model

Fractal theory was proposed by Mandelbrot and focused on studying the spatial structural complexity and fragmentation of objects [24]. A defining characteristic of fractal theory is self-similarity, which can manifest as local and global self-similarity, as well as generalized self-similarity [25]. Chen et al. further refined fractal theory and developed the C–A model [26], which was widely applied in identifying anomalies of geochemical element contents [11,27].

The C–A model is given by

$$A(c > \mu) \propto c^{-\lambda} \tag{1}$$

Here, $c$ represents geochemical element content, $\mu$ is a threshold, also named as anomaly lower limit of geochemical element content. $A(c > \mu)$ is an area where the $c$ is greater than the $\mu$, abbreviated as $A$, $\propto$ is a proportional relationship, $\lambda$ is a singularity index, or named a fractal dimension. The larger the $\mu$, the smaller the $A$, the changed degree of the $A$ is determined by $\lambda$.

In order to obtained some conclusions of the C–A model, (1) is rewritten as

$$A(c > \mu) = lc^{-\lambda} \tag{2}$$

Here, $l$ is a proportionality constant.

Taking natural logarithm of both sides of (2), we can obtain (3)

$$\ln A(c > \mu) = -\lambda \ln c + \ln l \tag{3}$$

In the double logarithmic coordinate $(\ln c, \ln A)$, there is a linear relationship between $\ln A$ and $\ln c$, and the slope of this linear relationship is the fractal dimension $\lambda$ of the C–A.

In the actual mining area, the distribution of geochemical element content follows a multifractal power-law distribution [11]. So, the area of geochemical element contents is a multicollinearity relationship in the $(\ln c, \ln A)$, which indicates that the area has multiple fractal dimensions. For this case, the least squares method is used to segmented linear fitting to form two linear lines. The intersection point of the two lines is the threshold $\mu$ of geochemical element contents, and the larger $\mu$, the smaller $A$.

## FA model

FA is a statistical method that can synthesize some variables with complex correlations into a few factors and can also obtain the dependency relationships between the variables. For a set of sample data $X = (x_1, x_2, \cdots, x_p)^T$, the FA model [21] is given by:

$$\begin{cases} x_1 = a_{11}F_1 + a_{12}F_2 + a_{1j}F_j + \cdots + a_{1m}F_m + \varepsilon_1, \\ x_2 = a_{21}F_1 + a_{22}F_2 + a_{2j}F_j + \cdots + a_{2m}F_m + \varepsilon_2, \\ \cdots\cdots \\ x_i = a_{i1}F_1 + a_{i2}F_2 + a_{ij}F_j + \cdots + a_{im}F_m + \varepsilon_i, \\ \cdots\cdots \\ x_p = a_{p1}F_1 + a_{p2}F_2 + a_{pj}F_j + \cdots + a_{pm}F_m + \varepsilon_p. \end{cases} \tag{4}$$

Here, $T$ is the transpose of a vector, $F_j$ is the jth common factor, $\varepsilon_i$ is the error, $a_{ij}$ is a factor loading, $i = 1, 2, \cdots, p$, $j = 1, 2, \cdots, m(m \leq p)$. (4) is the FA model of variables, called R-type factor analysis, the factor analysis for sample modeling is called Q-type factor analysis. We let

$$A = \begin{bmatrix} a_{11} & a_{12} & \cdots & a_{1m} \\ a_{21} & a_{22} & \cdots & a_{2m} \\ \vdots & \vdots & & \vdots \\ a_{p1} & a_{p2} & \cdots & a_{pm} \end{bmatrix}, X = (x_1, x_2, \cdots, x_m)^T, F = (F_1, F_2, \cdots, F_m)^T, \varepsilon = (\varepsilon_1, \varepsilon_2, \cdots, \varepsilon_P)^T$$

(4) is transformed into a matrix equation

$$X = AF + \varepsilon \tag{5}$$

Here, $A$ is a factor loading matrix. The FA needs to meet some properties [28]: ① $var(\varepsilon) = diag(\sigma_1^2, \sigma_2^2, \cdots, \sigma_m^2)$, $\sigma_i^2$ ($i = 1, 2, \cdots, p$) is a constant, $\varepsilon_1, \varepsilon_2, \cdots, \varepsilon_m$ are not related; ② $var(F) = I_m$, $I_m$ is an m-order identity matrix, $F_1, F_2, \cdots, F_m$ are not related; ③$cov(F, \varepsilon) = 0$, the $F$ and $\varepsilon$ are not related.

Now, we are about to determine the number of common factors and the factor loading matrix $A$. Hypothesizing that $R$ is the covariance of $X$, $\lambda_1, \lambda_2, \cdots, \lambda_p(\lambda_1 \geq \lambda_2 \geq \cdots \geq \lambda_p \geq 0)$ are characteristic roots of $R$ and $u_1, u_2, \cdots, u_p$ are the standardized orthogonal eigenvectors of $R$. Thus, $R$ is a real symmetric matrix and can undergo diagonalization. Then,

$$R = \sum_{i=1}^{p} \lambda_i u_i u_i^T = \left(\sqrt{\lambda_1}u_1, \sqrt{\lambda_2}u_2, \cdots, \sqrt{\lambda_p}u_p\right)\left(\sqrt{\lambda_1}u_1, \sqrt{\lambda_2}u_2, \cdots, \sqrt{\lambda_p}u_p\right)^T \tag{6}$$

Here, the value of error factor is 0, the number of variables is equal to the number of common factors. Eq. (6) is called the exact decomposition. In fact, it is essential that the number of common factors be less than the number of variables, so the decomposition of $R$ is:

$$R = \sum_{i=1}^{m} \lambda_i u_i u_i^T = \left(\sqrt{\lambda_1}u_1, \sqrt{\lambda_2}u_2, \cdots, \sqrt{\lambda_m}u_m\right)\left(\sqrt{\lambda_1}u_1, \sqrt{\lambda_2}u_2, \cdots, \sqrt{\lambda_m}u_m\right)^T, m \leq p \tag{7}$$

If we want to consider the effect of error factors, the covariance matrix $R$ is decomposed as follows

$$R = AA^T + \text{var}(\varepsilon) = \left(\sqrt{\lambda_1}\mu_1, \sqrt{\lambda_2}\mu_2, \cdots, \sqrt{\lambda_m}\mu_m\right)\left(\sqrt{\lambda_1}\mu_1, \sqrt{\lambda_2}\mu_2, \cdots, \sqrt{\lambda_m}\mu_m\right)^T + diag(\sigma_1^2, \sigma_2^2, \cdots, \sigma_m^2) \tag{8}$$

Thus, $A = (a_1, a_2, \cdots, a_m) = (a_1, a_2, \cdots, a_m) = \left(\sqrt{\lambda_1}u_1, \sqrt{\lambda_2}u_2, \cdots, \sqrt{\lambda_m}u_m\right)$. The number of common factors is estimated by the cumulative contribution rate $I, I = \sum_{i=1}^{m}\lambda_i / \sum_{i=1}^{p}\lambda_i$, $i = 1, 2, \cdots, p$, if $I > 80\%$, the number of common factors is $m$.

In the [Eq. (6)](), [Eq. (7)]() and [Eq. (8)](), if $R$ is unknown, we can use the sample variance of sample data instead of $R$. If the sample data has been standardized, the covariance $R$ is equal to the correlation coefficient matrix $M$ of $X$, $M = X^T X$. In addition, the standardized data can also eliminate the dimensional influence of variables. Therefore, the sample data need to be standardized before FA.

For a set of data $x_{ij}(i = 1, 2, \cdots, p, j = 1, 2, \cdots, m)$, the standardized method is given by [29]:

$$x'_{ij} = \frac{x_{ij} - \overline{x}_j}{\delta_j} \tag{9}$$

Here, $x'_{ij}$ is the standardized data for $x_{ij}$, $\overline{x}_j$, $\delta_j$ are the mean value and standard deviation of the ith variable $x_j$, respectively.

$$\text{cov}(x_i, F_j) = \text{cov}\left(\sum_{k=1}^{m} a_{ik}F_k + \varepsilon_j, F_j\right) = \text{cov}\left(\sum_{k=1}^{m} a_{ik}F_k, F_j\right) + \text{cov}(\varepsilon_j, F_j) = a_{ij} \tag{10}$$

As the sample data is the standardized data, the $D(x_i) = 1$, $D(F_j) = 1$, we can get

$$r_{x_i, F_j} = \frac{\text{cov}(x_i, F_j)}{\sqrt{D(x_i) D(F_j)}} = \text{cov}(x_i, F_j) = a_{ij} \tag{11}$$

So, the factor loading $a_{ij}$ is the covariance or correlation coefficient between the $x_i$ and $F_j$, indicating the degree to which $x_i$ depends on $F_j$. Therefore, in the factor loading matrix, the coefficients of the loading matrix for the same common factor are relatively large.

In the FA, we need to evaluate the role of each sample. So, we established a factor scoring model between common factors and variables as shown in (12).

$$F_j = b_{j1}x_1 + b_{j2}x_2 + \cdots b_{jm}x_m + b_j, \quad j = 1, 2, \cdots, m \tag{12}$$

Since the data has been standardized, $b_j = 0$.

Let $F = (F_1, F_2, \cdots, F_m)^T$, $X = (x_1, x_2, \cdots, x_m)$, $B = (b_{j1}, b_{j2}, \cdots, b_{jm})^T$, (12) is transformed into (13).

$$F = XB \tag{13}$$

Let $Q = (F - XB)^2$, according to the least squares method, we can obtain:

$$\begin{aligned}
\frac{\partial Q}{\partial B} &= \frac{(F-XB)^2}{\partial B} = \frac{\partial(F-XB)^T(F-XB)}{\partial B} \\
&= \frac{\partial(F^T - B^T X^T)(F-XB)}{\partial B} \\
&= \frac{\partial(F^T F - B^T X^T F - F^T XB + B^T X^T XB)}{\partial B} \\
&= \frac{\partial(F^T F - 2B^T X^T F + B^T X^T XB)}{\partial B} \\
&= -2X^T F + 2X^T XB = 0
\end{aligned} \tag{14}$$

Then, we can obtain (15) from (14)

$$B=(X^TX)^{-1}X^TF \tag{15}$$

From (5), it can be concluded that $A = XF^{-1}$. As $F$ is an orthogonal matrix, $F^{-1}=F^T, A = XF^{-1}=XF^T$. Thus,

$$F = XB=X(X^TX)^{-1}X^TF = X(M)^{-1}A^T \tag{16}$$

Here, $M = X^TX$.

## Designing FACA model

For a set of sample data $X = (x_{i1}, x_{i2}, \cdots, x_{ij} \cdots, x_{im}), i =1, 2, \cdots, p, j =1, 2, \cdots, m$, the different element combinations can be obtained by using the FA, the element combinations can determine the types of altered minerals. Using the C–A to identify geochemical anomalies of single elements, element combinations, and element comprehensive information, we can obtain the distribution law of these anomalies. Combining these anomaly distributions and metallogenic geological conditions, we can predict the location of minerals. Then a new FACA model based on the FA and C–A for predicting deep minerals is designed. The process of FACA is as follows:

**Step I:** Data standardization. Using (9) to standardize the sample data $X$, the standardized data does not influence the different magnitude and dimension.

**Step II:** Calculating correlation coefficient moment $M$. Calculating the $M$ of standardized data. $M = X^TX$.

**Step III:** Calculating eigenvalues and eigenvectors. The eigenvalues $\lambda_j$ and unitized orthogonal eigenvectors $u_j$ can obtain from $M$.

**Step IV:** Determine the number of common factors. The number of common factors is obtained by the cumulative contribution rate $I, I = \sum_{j=1}^{q} \lambda_j / \sum_{j=1}^{m} \lambda_j, j = 1, 2, \cdots, m$, if $I > 80\%$, the number of common factors is $q$. The common factor $F_j$ ($j = 1, 2, \cdots, q$) can reflect the combination information of different elements.

**Step V:** Factor Rotation. If the first $q$ common factors cannot clearly determine the different geochemical element combinations, we employ factor rotation to achieve more discernible element groupings. The objective of factor rotation lies in transforming the factor loadings' absolute values into a range closer to 0 and 1, thereby accentuating the differences between larger and smaller loadings. Factor rotation encompasses both orthogonal and oblique techniques, with this study utilizing the maximum variance orthogonal rotation method for factor transformation [30].

**Step VI:** Calculating factor score. To assess the significance of each factor, we compute the factor scores using Equation (16). When evaluating the comprehensive contribution ability of elements, we need to calculate the comprehensive score of the selected factors. The comprehensive score method is given by:

$$F = \sum_{j=1}^{q} w_j F_j \tag{17}$$

Here, $w_j = \frac{\lambda_j}{\lambda_1+\lambda_2+\cdots\lambda_q}$ ($j = 1, 2, \cdots, q$).

**Step VI:** Adjusting Geochemical Element Content Values. To prevent the clustering of geochemical elements in areas of high content and their sparsity in areas of low content, we process the element content values as outlined below:

$$c_j = I_{min} e^{\frac{j-1}{P} \ln\left(\frac{I_{max}}{I_{min}}\right)} \ (j = 1, 2, \cdots, m) \tag{18}$$

Here, $c_j$ is the processed value of geochemical element content, $P$ is the number of sampling points, $I_{min}$ is the unprocessed minimum value of geochemical element content, $I_{max}$ is the unprocessed maximum value of geochemical element content.

**Step VII:** Creating a double logarithmic scatter plot. In the $P$ sampling points, calculating the number of sampling points $p_j$ with element content value $c_k > \mu_j$, $k = 1, 2, \cdots, P$. Then, we draw a double logarithmic scatter plot $(\ln c_k, \ln N_j)$.

As the sampling is uniform, the number of sampling points $p_j$ is equivalent to the area $A$ where the $c_k > \mu_j$, the difference between the number of sampling points and the area $A$ is only a constant, which does not affect the curve shape in double logarithmic coordinates. So the $(\ln c_k, \ln N_j)$ is equivalent to the $(\ln c, \ln A)$.

**Step VIII:** Identifying anomalies of element content. In the double logarithmic plot, the intersection point $\mu_0$ of two straight lines is determined by using the least squares method, which named the anomaly lower limit of geochemical element content [11]. The values with $\mu_j > \mu_0$ called the anomalies of geochemical elements.

**Step IX:** Predicting the location of minerals. According to the coordinate position corresponding to anomalies, we can obtain the spatial distributions of anomalies of single elements, element combinations, and element comprehensive information by using MapGIS [31]. Combining these anomaly distributions and metallogenic geological conditions, we can predict the potential locations of minerals.

## Results

### The basic statistical results

FACA model was applied to predict deep minerals in the Jiguanzui copper gold mining area which was located at 114°54′42″–114°55′45″ E and 30°04′45″–30°05′50″ N. The study area explored more than 40 types of minerals, including iron, copper, lead, zinc, gold, silver, tungsten, molybdenum, gypsum, pyrite, limestone, dolomite, lapis lazuli, coal, perlite, bentonite, etc. The mining area is one of the important skarn-type iron copper mineralization areas in China. In the study area, the primary halo zoning of ore bodies is obvious which is mainly formed by the enrichment of geochemical element contents such as Ba, As, Ni, W, Au, Mo, Cu, and Ag. Ba and As are the leading edge halo elements, Ni and W are tail halo elements, and Au, Mo, Cu, and Ag are near-ore halo elements. At present, I, II, III, and VI main ore bodies have been discovered in the study area, and the ore body VII has enormous prospecting potential (Fig 1). Therefore, we selected the ore-forming elements Ba, As, Ni, W, Au, Mo, Cu, and Ag from 26 exploration lines of the ore body VII area for study. The sample data of geochemical element contents is shown in Table 1 and their basic statistical analysis is in Table 2. The complete data of Table 1 can be seen at https://github.com/qflong1/Studying-data.git (Table 1). The kurtosis and skewness values of geochemical element contents are greater than 3 and 0, respectively, so the element contents are distributed as a back tail distribution with no normal distribution. Using QQ plots [32] to perform skewness tests on these data (Fig 2), the element contents are skewed to the right. So, all the element contents are nonlinear distributions. We can use the new FACA model to process these data. With the depth of geology increasing, the element contents do not increase or decrease, they are changed in different forms (Fig 3), which indicates that there is a multi-stage hydrothermal ore-forming element superposition phenomenon. We believe that there should be a blind ore body in the deep part of the study area.

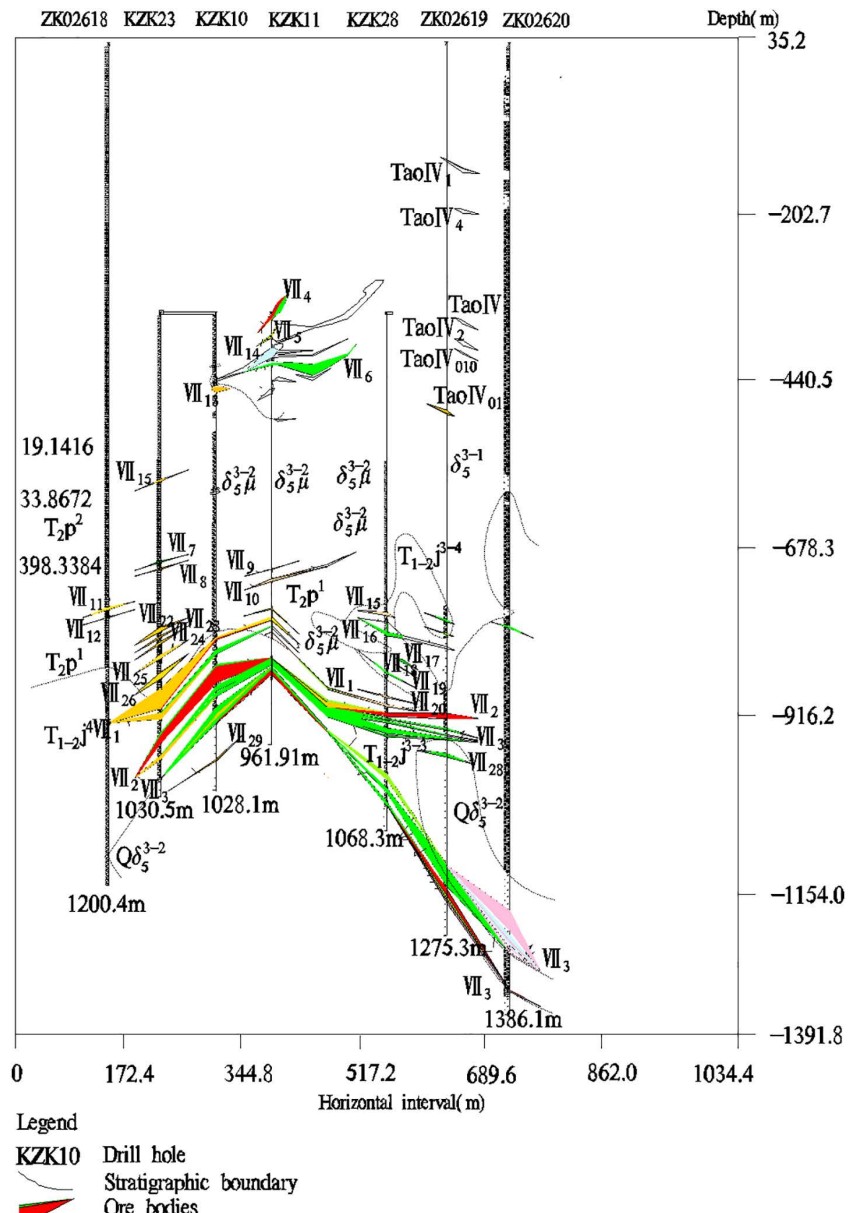

**Fig 1. The distribution pattern of ore body VII in the study area.** The horizontal axis represents the distance from the ground plane, and the vertical axis represents the distance from the surface to the depths of the Earth. The colored areas represent the distribution area of the ore body VII.

**Table 1. The data of geochemical element contents.**

| Sample No. | x | z | Ba | As | Ni | W | Au | Mo | Cu | Ag |
|---|---|---|---|---|---|---|---|---|---|---|
| 1 | 11.7047 | −1391.84 | 363.7042 | 37.0737 | 14.2818 | 6.141 | 39.1053 | 24.747 | 217.6846 | 0.4266 |
| 2 | 13 | −1391.84 | 358.0914 | 36.2547 | 14.7306 | 6.4208 | 39.2186 | 24.417 | 214.5438 | 0.4185 |
| 3 | 16.1555 | −1391.84 | 353.9632 | 35.4079 | 15.0631 | 6.9182 | 39.4019 | 23.96 | 212.3366 | 0.4103 |
| … | … | … | … | … | … | … | … | … | … | … |
| 36717 | 891.709 | 35.16 | 461.7822 | 59.9851 | 11.3214 | 4.6036 | 88.4197 | 5.567 | 290.5749 | 0.2629 |

**Table 2. Statistical analysis of elements.**

| Elements | Mean | Minimum | Maximum | median | Std. Dev | Covariance | Skewness | Kurtosis |
|---|---|---|---|---|---|---|---|---|
| Ba | 356.7463 | 11.0948 | 1618.3355 | 356.771 | 137.5827 | 18929.0093 | 4.6494 | 4.6946 |
| As | 19.4261 | 2.3788 | 238.1399 | 14.5565 | 13.8788 | 192.6210 | 1.8643 | 5.6106 |
| Ni | 18.4461 | 3.7045 | 111.6446 | 16.7013 | 7.7957 | 60.7723 | 1.5813 | 4.3233 |
| W | 12.8465 | 1.1572 | 350.8949 | 12.006 | 8.8482 | 3457.3732 | 9.3684 | 246.5948 |
| Au | 70.2634 | 5.1396 | 2813.3192 | 59.7161 | 58.7994 | 3457.3731 | 11.3245 | 319.6122 |
| Mo | 23.2319 | 2.0500 | 474.4456 | 19.1417 | 18.9687 | 359.8119 | 4.4173 | 49.2166 |
| Cu | 625.0878 | 37.5401 | 30952.5169 | 387.9453 | 737.8724 | 544455.6408 | 10.3767 | 237.3541 |
| Ag | 0.491 | 0.0523 | 3.70315 | 0.4554 | 0.2478 | 0.0614 | 20.392 | 11.752 |

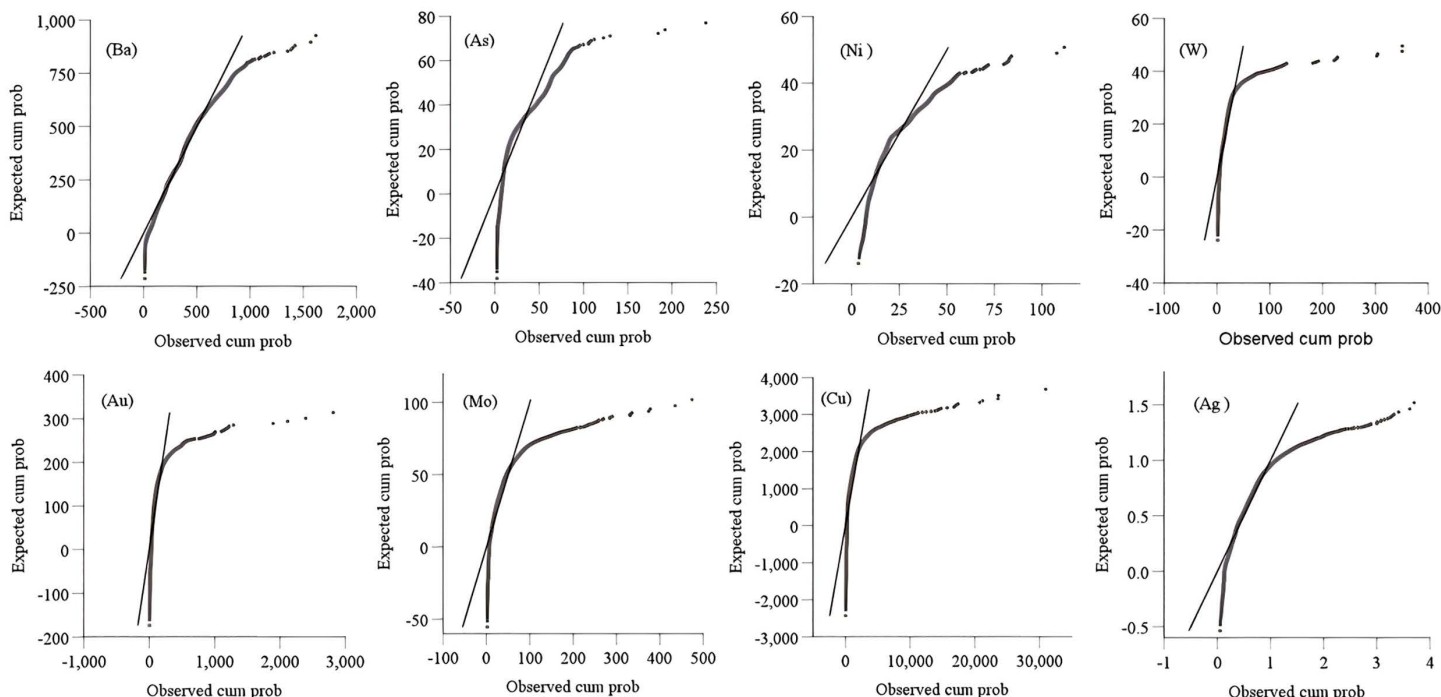

**Fig 2. Skewness tests through Q-Q plots.** The content of 8 elements is offset to the right of the straight line, which does not satisfy the linear relationship and has nonlinear characteristics.

## Determining the combinations of geochemical elements

Here, we used the designed FACA model to process and analyze the study data. The standardized data (Table 3) were obtained using (9) to process the data of Table 1. The complete data of Table 3 can be seen at https://github.com/qflong1/Studying-data.git (Table 3). In Table 3, the standardized data for each element is relatively concentrated, and the dimensional influence of the original data has been removed. Calculating the correlation coefficient matrix of Table 3 is shown in Table 4, the value of the correlation coefficient of Ag, Au, Mo, and Cu is relatively high, which indicates that these elements have strong correlations, it is consistent with they are near ore halo elements. The correlation coefficients between Ni and Ba, Ni and As are 0.3566 and −0.2936, respectively, the results show the coexistence of the leading edge and tail halo elements, indicating a new ore body in the deep earth.

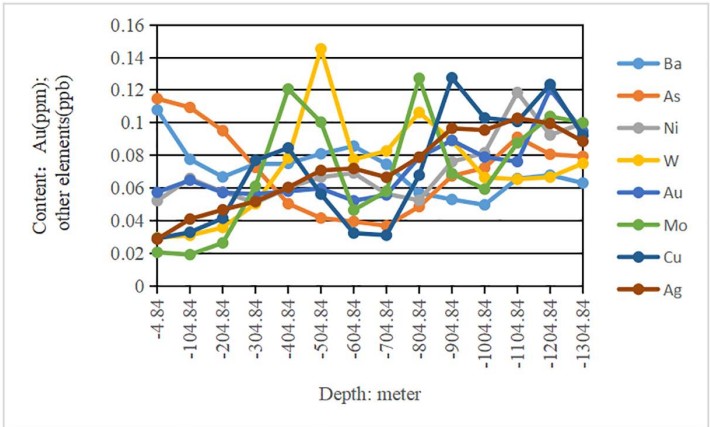

**Fig 3. Relationship between the element contents and different depths.** Different colors represent different geochemical elements.

**Table 3. Standardized data for elements.**

| Sample No. | x | z | Ba | As | Ni | W | Au | Mo | Cu | Ag |
|---|---|---|---|---|---|---|---|---|---|---|
| 1 | 11.7047 | −1391.84 | 0.0506 | 1.2716 | −0.5342 | −0.7578 | −0.5299 | 0.0799 | −0.5521 | −0.2596 |
| 2 | 13 | −1391.84 | 0.0098 | 1.2125 | −0.4766 | −0.7262 | −0.528 | 0.0625 | −0.5564 | −0.2923 |
| 3 | 16.1555 | −1391.84 | −0.0202 | 1.1515 | −0.434 | −0.67 | −0.5249 | 0.0384 | −0.5594 | −0.3254 |
| … | … | … | … | … | … | … | … | … | … | … |
| 36717 | 891.709 | 35.16 | 0.0506 | 1.2716 | −0.5342 | −0.7578 | −0.5299 | 0.0799 | −0.5521 | −0.2596 |

The eigenvectors and eigenvalues of Table 4 calculated are shown in Table 5. The cumulative contribution rate of the eigenvalues of the first four common factors was 78.32%, close to 80%. So, the first four common factors to extract the comprehensive information met the requirement of FA. The factor loading matrix was obtained as shown in Table 6, but the factor loadings were relatively close, so it wasn't easy to select element combinations. Therefore, the factor loading matrix (Table 6) needed to factor rotation, and the rotated load matrix (Table 7) was obtained using orthogonal rotation. In Table 7, from the first common factor F1, the factor loadings of W and Mo were 0.881 and 0.8321, respectively, relatively high, W and Mo are considered a combination of altered minerals. In F2, the factor loadings of Ba and Ni were 0.9103 and 0.6835, respectively, relatively high, indicating that Ba and Ni have strong contribution ability, they are considered a combination. The coexistence phenomenon of the leading-edge halo element Ba and the tail halo

**Table 4. Correlation coefficient matrixes of elements.**

| | Ba | As | Ni | W | Au | Mo | Cu | Ag |
|---|---|---|---|---|---|---|---|---|
| Ba | 1 | 0.1226 | 0.3566 | 0.0991 | 0.0242 | 0.006 | −0.0558 | −0.2035 |
| As | 0.1226 | 1 | −0.2936 | −0.1339 | −0.0134 | −0.1403 | −0.0155 | 0.1367 |
| Ni | 0.3566 | −0.2936 | 1 | 0.1623 | −0.1748 | −0.0746 | −0.1471 | −0.1377 |
| W | 0.0991 | −0.1339 | 0.1623 | 1 | 0.0736 | 0.5954 | 0.2822 | 0.4271 |
| Au | 0.0242 | −0.0134 | −0.1748 | 0.0736 | 1 | 0.1725 | 0.3937 | 0.4158 |
| Mo | 0.006 | −0.1403 | −0.0746 | 0.5954 | 0.1725 | 1 | 0.334 | 0.4111 |
| Cu | −0.0558 | −0.0155 | −0.1471 | 0.2822 | 0.3937 | 0.334 | 1 | 0.6629 |
| Ag | −0.2035 | 0.1367 | −0.1377 | 0.4271 | 0.4158 | 0.4111 | 0.6629 | 1 |

**Table 5. The eigenvectors and eigenvalues of correlation coefficient.**

|  | $\mu_1$ | $\mu_2$ | $\mu_3$ | $\mu_4$ | $\mu_5$ | $\mu_6$ | $\mu_7$ | $\mu_8$ |
|---|---|---|---|---|---|---|---|---|
| Unitized orthogonal eigenvector | −0.0838 | −0.4034 | −0.716 | −0.0794 | 0.2704 | 0.3349 | −0.113 | 0.3365 |
|  | −0.0167 | 0.3963 | −0.6324 | 0.4701 | −0.1884 | −0.2401 | 0.1663 | −0.3177 |
|  | −0.1251 | −0.62 | −0.1017 | −0.2571 | −0.4945 | −0.2992 | 0.3679 | −0.2327 |
|  | 0.3934 | −0.4213 | 0.0332 | 0.3695 | 0.01 | −0.27 | −0.6444 | −0.2048 |
|  | 0.3395 | 0.1773 | −0.2166 | −0.6215 | 0.4156 | −0.4575 | −0.0163 | −0.1944 |
|  | 0.4337 | 0.2519 | 0.1289 | 0.3537 | 0.4424 | 0.0932 | 0.6301 | −0.0731 |
|  | 0.485 | 0.0926 | −0.0895 | −0.2422 | −0.3304 | 0.6419 | −0.0527 | −0.4069 |
|  | 0.5326 | 0.1223 | −0.0662 | −0.0023 | −0.4112 | −0.1944 | 0.0951 | 0.6936 |
| Eigenvalues | $\lambda_1$ | $\lambda_2$ | $\lambda_3$ | $\lambda_4$ | $\lambda_5$ | $\lambda_6$ | $\lambda_7$ | $\lambda_8$ |
|  | 2.5791 | 1.5988 | 1.1348 | 0.954 | 0.6907 | 0.4937 | 0.339 | 0.21 |

**Table 6. Load matrix of common factors.**

|  | F1 | F2 | F3 | F4 |
|---|---|---|---|---|
| Ba | −0.1345 | 0.51 | 0.7627 | 0.0775 |
| As | −0.0268 | −0.5011 | 0.6737 | −0.4591 |
| Ni | −0.2009 | 0.7839 | 0.1083 | 0.2511 |
| W | 0.6318 | 0.5326 | −0.0354 | −0.3609 |
| Au | 0.5453 | −0.2242 | 0.2307 | 0.607 |
| Mo | 0.6966 | 0.3185 | −0.1373 | −0.3455 |
| Cu | 0.7789 | −0.1171 | 0.0953 | 0.2366 |
| Ag | 0.8553 | −0.1546 | 0.0705 | 0.0023 |

**Table 7. The rotated load matrix.**

|  | F1 | F2 | F3 | F4 |
|---|---|---|---|---|
| Ba | 0.0049 | 0.9103 | 0.1932 | 0.0057 |
| As | −0.0742 | 0.0467 | 0.9533 | −0.0014 |
| Ni | 0.0663 | 0.6835 | −0.4793 | −0.1683 |
| W | 0.881 | 0.1581 | −0.088 | 0.0743 |
| Au | −0.0867 | 0.0411 | −0.0343 | 0.8712 |
| Mo | 0.8321 | −0.0639 | −0.0495 | 0.1609 |
| Cu | 0.3432 | −0.0926 | 0.0254 | 0.7473 |
| Ag | 0.5107 | −0.1938 | 0.1562 | 0.6616 |

element Ni indicated that there may be a new ore body in the study area. In F3, the factor loading of As was 0.9533, the largest factor loading, it has a strong contribution ability and is a separate category. In F4, the factor loadings of Au, Cu, and Ag were 0.8712, 0.7473, and 0.6616, respectively, relatively high, the contribution ability of Au, Cu, and Ag are strong and are considered as a combination of altered minerals. So, the elements in the study area were divided into four combinations: {W and Mo}, {Ba and Ni}, {As}, and {Au, Cu, and Ag}. The four element combinations were obtained by using FA that conformed to the classification criteria of primary halo geochemical elements, which was beneficial as a basis for mineral prediction.

## Identifying anomalies of geochemical element contents

When the factor scores of four common factors were obtained by using (16) and the comprehensive scores of the four common factors were obtained by using (17) (Table 8). The complete data of Table 8 can be seen at https://github.com/qflong1/Studying-data.git (Table 8). The C–A will be used to identify anomalies of geochemical elements and their combinations. Using FA, we obtained the double logarithmic scatter plots of Ba, As, Ni, W, Au, Mo, Cu, and Ag as shown in Fig 4. The thresholds of the 8 elements were obtained by using the double logarithmic scatter plots as shown in Table 9. To verify whether the thresholds determined by FA were reasonable, diagnostic tests were conducted on the rationality of selection thresholds by using P-P plots [32] as shown in Fig 5. The actual distributions of all element contents exceeding their thresholds tended towards a straight line of theoretical distributions, indicating that the established C–A model was reasonable for identifying thresholds of geochemical element contents. Similarly, we obtained the double logarithmic scatter plots (Fig 6) and the thresholds (Table 10) of the factor scores of four common factors and their comprehensive factor, these thresholds also met the diagnostic tests (Fig 7). Thus, we identified the anomaly lower limits of Ba, As, Ni, W, Au, Mo, Cu, and Ag were 398.3384, 25.3168, 19.1416, 16.7271, 79.7408, 28.7244,

**Table 8. Scores and comprehensives score of common factors.**

| Sample No. | x | z | F1 | F2 | F3 | F4 | F |
|---|---|---|---|---|---|---|---|
| 1 | 11.7047 | −1391.84 | −0.2227 | −0.2006 | 1.2712 | −0.6056 | −0.0048 |
| 2 | 13 | −1391.84 | −0.2221 | −0.2028 | 1.191 | −0.6176 | −0.0215 |
| 3 | 16.1555 | −1391.84 | −0.2124 | −0.2017 | 1.1167 | −0.6303 | −0.0326 |
| … | … | … | … | … | … | … | … |
| 36717 | −1.1133 | 0.5283 | 2.7155 | −0.1343 | −1.1133 | 0.5283 | 0.14788 |

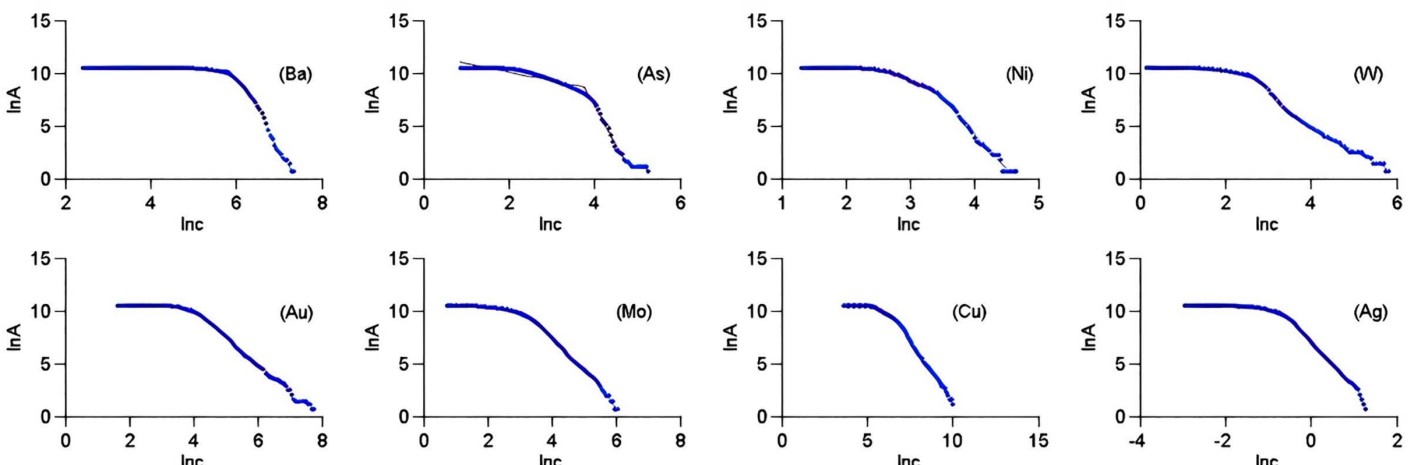

**Fig 4. Double logarithmic scatter plots of geochemical element contents.**

**Table 9. Thresholds of geochemical elements.**

| Elements | Ba | As | Ni | W | Au | Mo | Cu | Ag |
|---|---|---|---|---|---|---|---|---|
| Threshold | 398.3384 | 25.3168 | 19.1416 | 16.7271 | 79.7408 | 28.7244 | 896.1419 | 0.6393 |

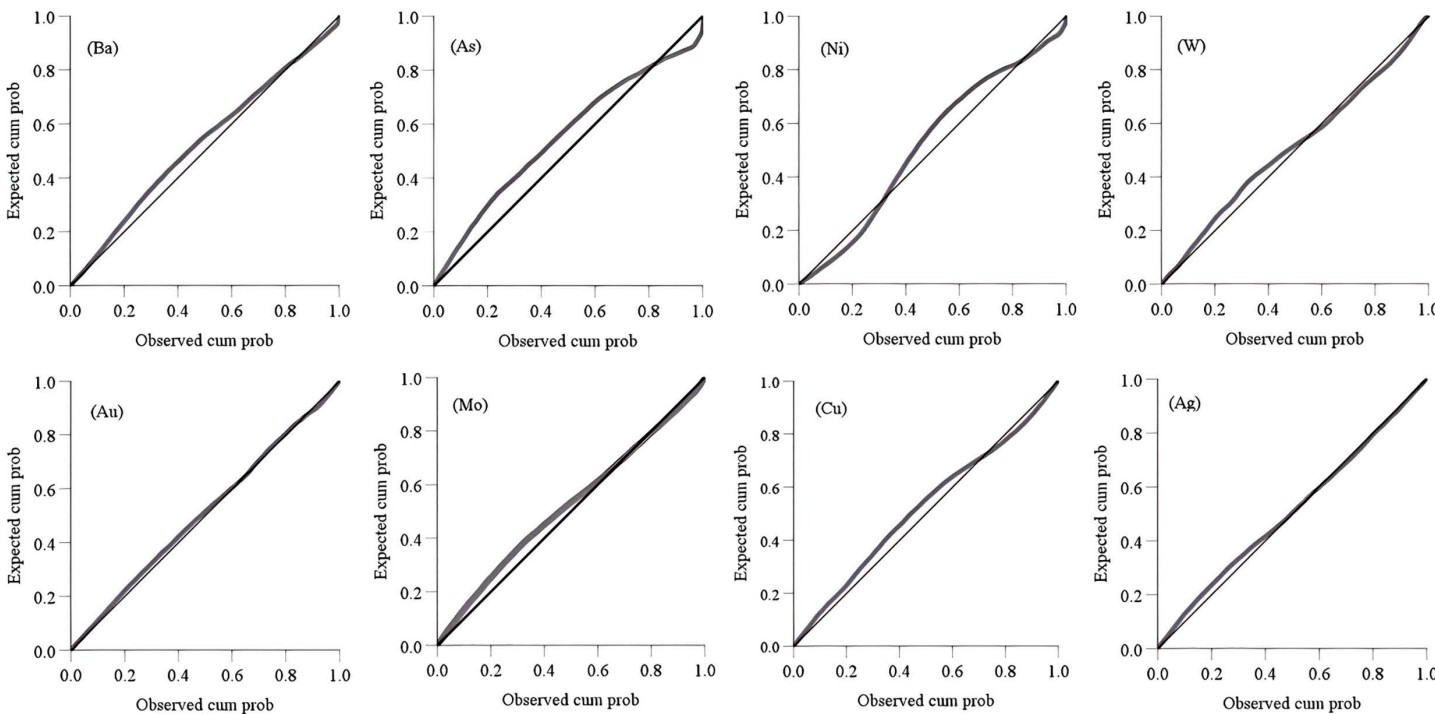

**Fig 5. Diagnostic tests for thresholds of geochemical elements.** The curve represents the actual distribution of data with element content greater than their anomaly lower limits, while the straight line represents the theoretical distribution of data with element content greater than their anomaly lower limit.

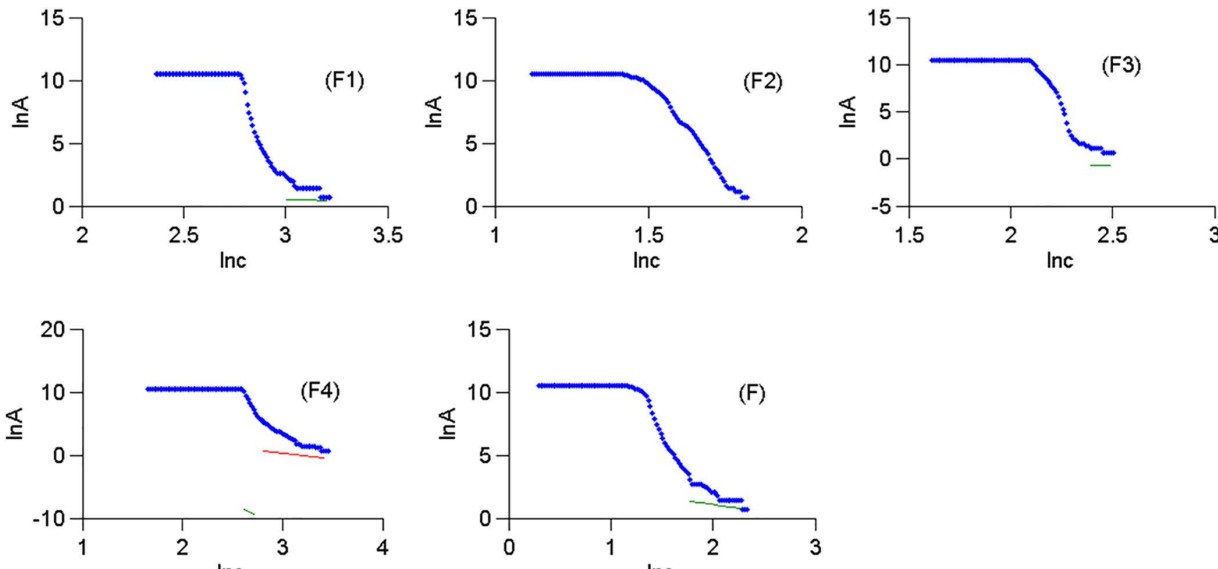

**Fig 6. Double logarithmic scatter plots of factors.** (F1-F4) The first four common factors, F is the comprehensive factor.

**Table 10. Threshold of factors.**

| Factors | F1 | F2 | F3 | F4 | F |
|---|---|---|---|---|---|
| Threshold | 0.4321 | 0.6363 | 0.3943 | 0.2168 | 0.2958 |

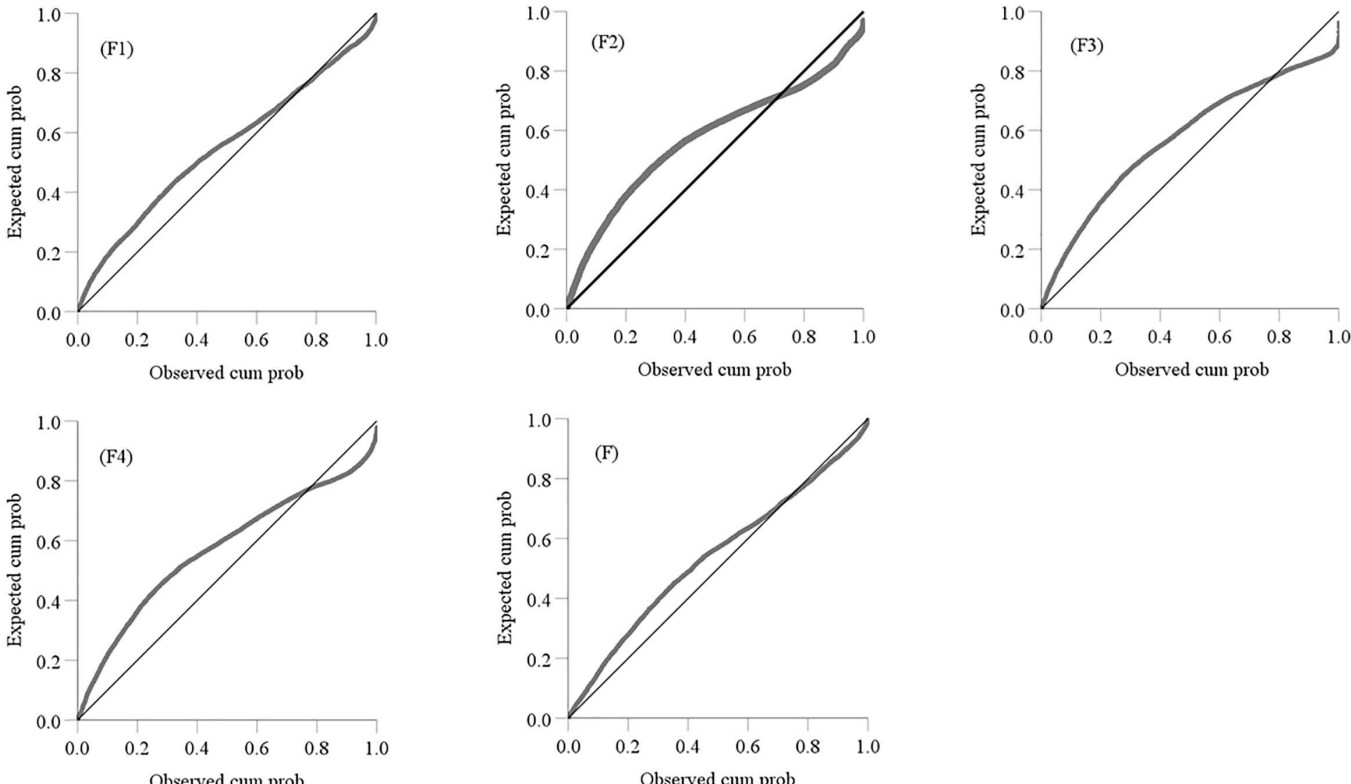

**Fig 7. Diagnostic tests for thresholds of factors.** (F1-F4) The first four common factors, F is the comprehensive factor. The curve represents the actual distribution of factor data greater than their anomaly lower limits, while the straight line represents the theoretical distribution of factor data greater than their anomaly lower limits.

896.1419, and 0.6393, respectively, the thresholds of factor scores for {W and Mo}, {Ba and Ni}, {As}, and {Au, Cu, and Ag} were 0.4321, 0.6363, 0.3943, and 0.2958, respectively, and the threshold of the comprehensive information of the 8 elements was 0.2958.

## Distribution law of geochemical elements and mineral prediction

When the thresholds for elements and their combinations had been estimated, the geochemical element contents beyond their thresholds were called the anomaly values. Considering the anomaly values of geochemical elements and the coordinate positions of anomaly values, we can obtain the spatial distributions of geochemical elements by MapGIS [31] as shown in Figs 8 and 9. The leading edge halo elements Ba and As were not only enriched in the upper parts of ore body VII but also enriched in the lower parts of ore body VII (Fig 8). The tail halo element W was enriched in the lower and upper parts of ore body VII (Fig 8), this is an abnormal reverse zoning of the leading edge halo element and tail halo element, there may be some new ore bodies in the deep study area. The near-ore halo elements Au, Mo, Cu, and Ag

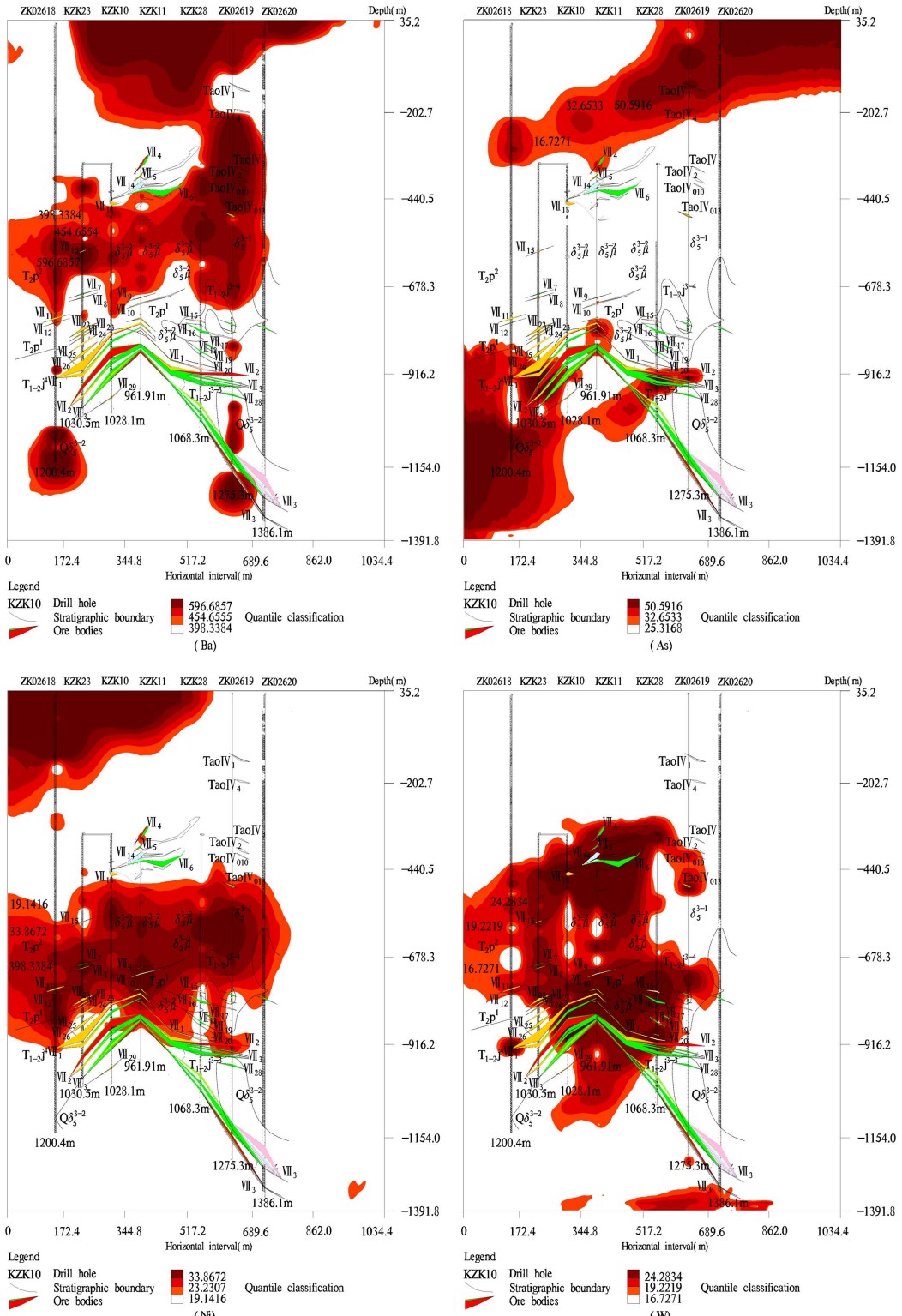

**Fig 8. Abnormal distribution law of the content of leading halo elements and tail halo elements.** The red area of each element represents the abnormal distribution area of element content.

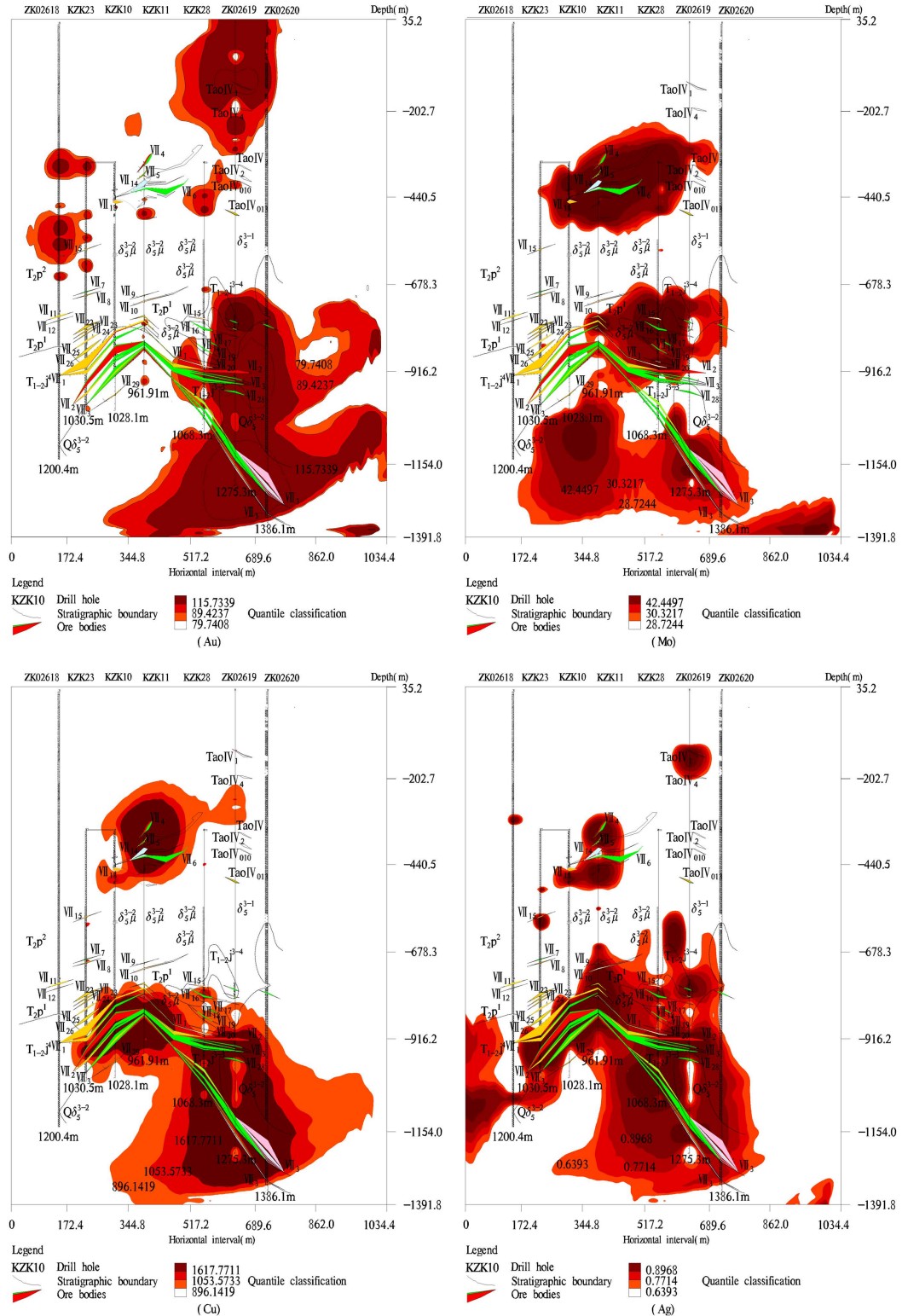

**Fig 9. Abnormal distribution law of the content of leading edge halo elements.** The red area of each element represents the abnormal distribution area of element content.

were strongly enriched near the ore body VII (Fig 9), these anomaly areas of near ore halo elements are consistent with the actual distribution of ore bodies, which shows that the designed C–A model of this paper is reasonable. Specifically, Mo had strong anomalous phenomena in the lower parts of ore body VII, it is shown that there may be a molybdenum ore body in the deep areas.

To predict the specific location of the new ore body, we obtained the spatial distribution laws (Fig 10) of four common factor scores and comprehensive distribution law (Fig 11(a)) of four common factor comprehensive scores. From Fig 10, the element combinations {W and Mo} of the first common factor were enriched in the lower and upper parts of ore body VII. Especially, Mo had a strong enrichment phenomenon below ore body VII, demonstrating that the molybdenum ore body may be extended downwards. The combinations {Ba and Ni} of the second common factor were enriched simultaneously on the ore body VII, it was a coexistence phenomenon of the front halo element and tail halo element, indicating that there may be a new ore body in the deep areas. The third common factor {As} was not only enriched above the ore body VII, but also strongly enriched below the ore body VII, it belonged to the anti-anomaly distribution of the leading edge halo element. So, we further believe that there is a new ore body in the study area. The element combinations {Au, Cu, Ag} of the fourth common factor were mainly enriched on the ore body VII, which indicates that the near ore halo elements are the main prospecting indicator of ore bodies. Therefore, the distribution laws of four element combinations are consistent with the properties of near ore halo elements, near ore halo elements, and tail halo elements, which proves the designed FACA model is reasonable.

In addition, the comprehensive anomaly distribution of the four common factors was shown in Fig 11(a), which covered all the ore body VII, it is shown that the comprehensive anomaly distribution law of geochemical elements has an important indicative role in the mineral prediction. From the depth of about 1000 m below the drill hole ZK02619 to the depth of about 1170 m below the drill hole ZK02620, the comprehensive anomaly of elements has strong enrichment, indicating the existence of a blind ore body at this location, and we have discovered molybdenum ore and copper-gold deposits at this location by geological prospecting [33], which proved the designed FACA model of the paper can be used for actual mineral exploration. On the left side of ore body VII, there is a strong enrichment of the element comprehensive at a depth of about 1120m to 1150m between drill holes ZK02618 and KZK23, there may be a blind ore body. The anomaly of near-ore halo element Mo also has strong enrichment at the location, further explaining the existence of blind ore bodies. The third common factor {As} was also enriched above and below the ore body VII, it belonged to the anti-anomaly distribution of the leading edge halo element. This phenomenon indicates there is a new ore body in the study area. Combined with the trend of ore body VII, there should be a molybdenum blind ore body at the location. The new ore body may be a left extension of the middle part of ore body VII where at a depth of about 800 m to 1050 m below drill holes KZK23, KZK10, and KZK28. Therefore, we have predicted there is a new molybdenum blind ore body at a depth of about 1120 m to 1150m below drill holes ZK02618 and KZK23, which is shown in Fig 11(b).

## Uncertainty and risk analysis

The successful prediction of mineral resources can effectively search for minerals, but no prediction has a 100% success rate. If the prediction accuracy of mineral location is not high, it is impossible to survey the ore body. Mineral exploration involves the cost of road construction, wear and tear of drilling tools, machine handling, and so on. Mineral prediction plays an important part in mineral exploration. To verify the accuracy of the mineral prediction model, the designed model was validated through actual exploration. From a depth of approximately 1000m below borehole ZK02619 to a depth of approximately 1170m below borehole ZK02620, there is a strong anomalous enrichment of near-ore halo elements Au, Mo, Cu, and Ag. The {Au, Cu, and Ag} element combination exhibits a strong enrichment phenomenon, and the comprehensive anomaly of elements is also strongly enriched, indicating the possible existence of blind ore bodies at this location. The 2016 exploration program carried out by the First Geological Brigade of the Hubei Geological Bureau revealed the presence of molybdenum ore and copper-gold deposits at the site, as evidenced by the final borehole ZK02620 [33]. The success of actual mineral exploration proved the rationality of the designed FACA model. On the left side of ore body

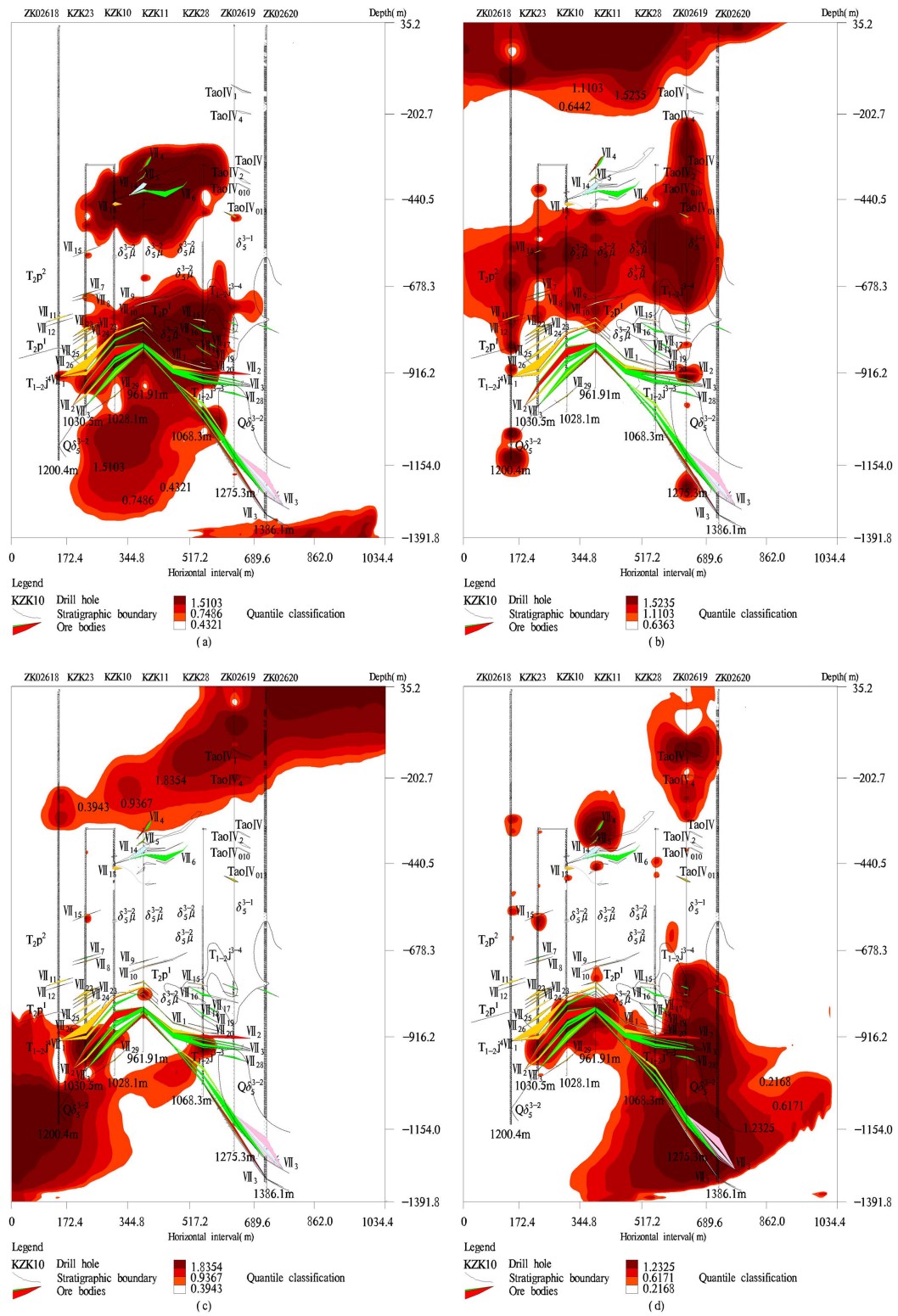

**Fig 10. Anomaly distribution law of the four common factors.** (a-d) the first four common factors.

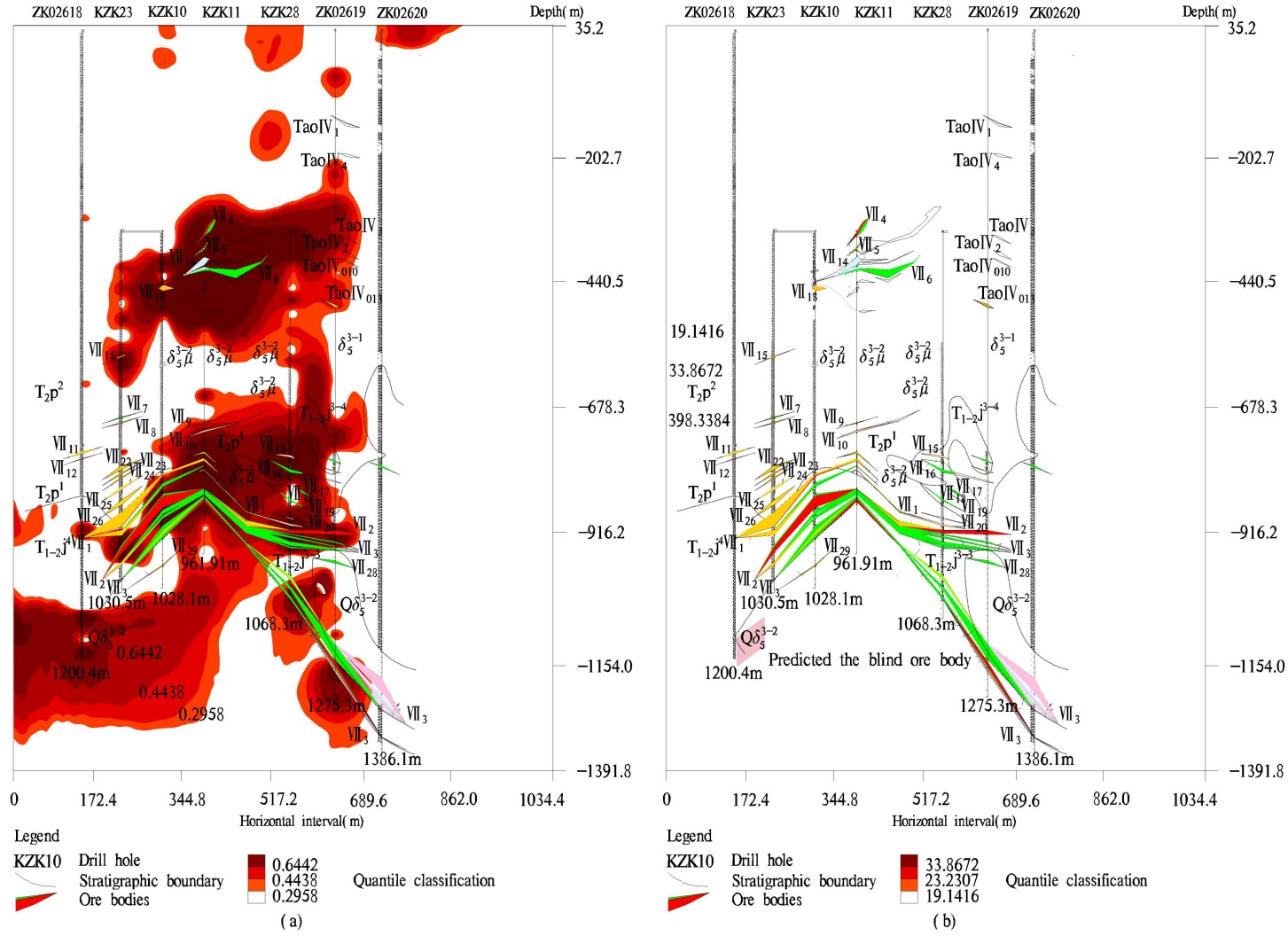

**Fig 11. Ore body prediction. (a)** Comprehensive anomaly law of all elements, **(b)** Predicted the new molybdenum blind ore body. The pink small parallelogram area of (b) at a depth of approximately 1120 to 1150 meters below boreholes ZK02618 and KZK23 is the new Predicted ore body area.

VII, the third common factor {As} was also enriched above and below the ore body VII, it belonged to the anti-anomaly distribution of the leading edge halo element. This phenomenon indicates there is a new ore body in the study area. The element comprehensive has a strong enrichment phenomenon at a depth of about 1120m to 1150m between drill holes ZK0268 and KZK23. Near-ore halo element Mo also has strong enrichment at this location, indicating there may be a blind ore body. Combined with the trend of ore body VII, there should be a molybdenum blind ore body. According to the actual exploration verification results on the right side of ore body VII, the location of the new blind ore body predicted on the left side of ore body VII using the same method is reasonable, which is beneficial for later mineral exploration.

## Conclusions

In this study, considering the nonlinear characteristics of the distribution of geochemical element contents and mineral resources formed by multiple element combinations, we established a C–A model for geochemical anomaly extraction and

FA was used to obtain element combinations. On this basis, a new FACA model for mineral prediction was designed and successfully applied to actual geological mineral prediction. The main conclusions of this study were as follows:

(1) According to the nonlinear characteristics of the distribution of geochemical element contents, the C–A model was designed for identifying geochemical anomalies. The threshold of the C–A model was estimated by creating a double logarithmic scatter plot. Through diagnostic testing, the theoretical distribution was consistent with the actual distribution of geochemical element contents beyond their thresholds, demonstrating that the designed C–A model was effective.

(2) An FA model was established for extracting the different combinations and comprehensive information of geological elements. Based on the C–A and FA models, FACA model was designed for mineral prediction, and the parameters of FACA were estimated. By processing geochemical elements using the FACA model, we obtained different altered mineral combinations: {W and Mo}, {Ba and Ni}, {As}, and {Au, Cu, and Ag} by using the FACA model and various prospecting indicators.

(3) FACA model was applied to the study area for mineral prediction. The results showed that the comprehensive abnormality distribution law of elements was consistent with the trend of the ore body VII. The front halo elements and tail halo elements had a coexisting relationship in spatial distribution, there was a blind ore body in the deep area. The comprehensive information of geochemical elements had strong anomaly distribution below ore body VII. Here, Mo also had a strong enrichment phenomenon. Considering the geological condition, the trend of the ore body, and the anomaly distribution of element combinations, we predicted there was a new blind ore body located at a depth of about 1120m - 1150m below ground between drill holes ZK02618 and KZK23.

## Acknowledgments

The authors would like to thank the research team of the Key Laboratory of Mathematical Geology in Sichuan, China.

## Author contributions

**Conceptualization:** Feilong Qin.

**Formal analysis:** Yu Feng.

**Funding acquisition:** Feilong Qin.

**Investigation:** Feilong Qin, Hongjin Zhu, Yu Feng.

**Methodology:** Feilong Qin, Hongjin Zhu, Yu Feng.

**Project administration:** Feilong Qin.

**Supervision:** Feilong Qin.

**Writing – original draft:** Feilong Qin, Hongjin Zhu, ShiCheng Yu.

**Writing – review & editing:** Feilong Qin.

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
