## [Decision Letter · Decision Letter 0]

16 Sep 2025

PONE-D-25-39619Using fractal model and factor analysis for FACA modeling and its application in deep mineral predictionPLOS ONE

Dear Dr. Qin,

Thank you for submitting your manuscript to PLOS ONE. After careful consideration, we feel that it has merit but does not fully meet PLOS ONE’s publication criteria as it currently stands. Therefore, we invite you to submit a revised version of the manuscript that addresses the points raised during the review process.

We look forward to receiving your revised manuscript.

Kind regards,

Hu Li

Academic Editor

PLOS ONE

Journal Requirements:

1. Please ensure that your manuscript meets PLOS ONE’s style requirements, including those for file naming. The PLOS ONE style templates can be found at

“The authors are grateful to the Hubei Geological Bureau for providing the study data. This study was supported by the Opening Fund of Geomathematics Key Laboratory of Sichuan Province (Grant no. scsxdz2022-06), and the Program of Sichuan Mineral Resources Research Center (Grant no.  SCKCZY2023-ZC006).”

“All of the authors have read the journal’s policy and also had no conflict of interests.”

6. Please note that your Data Availability Statement is currently missing the the DOI/accession number of each dataset or a direct link to access each database. If your manuscript is accepted for publication, you will be asked to provide these details on a very short timeline. We therefore suggest that you provide this information now, though we will not hold up the peer review process if you are unable.

7. When completing the data availability statement of the submission form, you indicated that you will make your data available on acceptance. We strongly recommend all authors decide on a data sharing plan before acceptance, as the process can be lengthy and hold up publication timelines. Please note that, though access restrictions are acceptable now, your entire data will need to be made freely accessible if your manuscript is accepted for publication. This policy applies to all data except where public deposition would breach compliance with the protocol approved by your research ethics board. If you are unable to adhere to our open data policy, please kindly revise your statement to explain your reasoning and we will seek the editor’s input on an exemption. Please be assured that, once you have provided your new statement, the assessment of your exemption will not hold up the peer review process.

8. We notice that your supplementary figures and tables are included in the manuscript file. Please remove them and upload them with the file type ’supporting Information'. Please ensure that each Supporting Information file has a legend listed in the manuscript after the references list.

Reviewers' comments:

Reviewer’s Responses to Questions

**Comments to the Author**

1. Is the manuscript technically sound, and do the data support the conclusions?

Reviewer #1: Yes

Reviewer #2: Yes

2. Has the statistical analysis been performed appropriately and rigorously? 

Reviewer #1: N/A

Reviewer #2: Yes

3. Have the authors made all data underlying the findings in their manuscript fully available?

Reviewer #1: No

Reviewer #2: Yes

4. Is the manuscript presented in an intelligible fashion and written in standard English?

Reviewer #1: Yes

Reviewer #2: Yes

5. Review Comments to the Author

Reviewer #1: The manuscript “Using fractal model and factor analysis for FACA modeling and its application in deep mineral prediction” presents a combined FACA approach for geochemical anomaly extraction and mineral prediction. The study is well-motivated, the methodology is clearly described, and the case study in the Jiguanzui mining area provides convincing evidence of the effectiveness of the proposed model.

The integration of the concentration–area (C–A) model with factor analysis (FA) is appropriate and novel in application. The results are consistent with known ore bodies and provide plausible predictions of potential blind ore bodies. The conclusions are supported by the evidence, and the manuscript is overall well-structured and informative.

One important issue concerns data availability. The current statement indicates that data may be obtained from the corresponding author upon request. This arrangement does not fully comply with the PLOS ONE Data Policy, which requires deposition in an open and unrestricted public repository. I encourage the authors to deposit the relevant datasets and analysis scripts in a recognized repository (e.g., Zenodo, OSF, Figshare) and provide persistent links in the revised manuscript to ensure compliance.

Overall, the work is technically sound and suitable for publication, provided that the data availability issue is addressed. I recommend acceptance pending minor revision to ensure full compliance with data-sharing requirements.

Reviewer #2: The paper combines the C - A model and the FA method to design a new model FACA, which is used to determine the anomaly values of geochemical elements, thereby improving the accuracy of deep mineral prediction. The paper selects a representative specific study area to verify the application effect of the model. The processing of geochemical data in the study area is rigorous, and the conclusions are consistent with the actual situation of the study area, which can guide further mineral resource exploration in the study area. The paper has a clear logic, an innovative model, and supported conclusions. It is recommended to accept this paper. However, some language expressions in the paper need further correction. For example, in line 18, "model named the FACA for mineral prediction" is suggested to delete "the".

6. PLOS authors have the option to publish the peer review history of their article (what does this mean?). If published, this will include your full peer review and any attached files.

Reviewer #1: No

Reviewer #2: No

---

## [Editor Report · Decision Letter 1]

28 Nov 2025

PONE-D-25-39619R1Using fractal model and factor analysis for FACA modeling and its application in deep mineral predictionPLOS ONE

Dear Dr. Qin,

Thank you for submitting your manuscript to PLOS ONE. After careful consideration, we feel that it has merit but does not fully meet PLOS ONE’s publication criteria as it currently stands. Therefore, we invite you to submit a revised version of the manuscript that addresses the points raised during the review process.

We look forward to receiving your revised manuscript.

Kind regards,

Hu Li

Academic Editor

PLOS ONE

Journal Requirements:

Additional Editor Comments:

The reviewers have confirmed the acceptance of this paper. However, there are some minor issues that need to be addressed before publication:

1. I did not see the abstract of the paper, which is an indispensable component. The abstract should specifically highlight the fractal methodology used in this study and its application results.

2. Significant research has been conducted on fractal theory in deep mineral exploration and development, which is not reflected in the Introduction. It is essential to include the latest advancements related to fractal theory and mineral exploration, with reference to works such as: https://doi.org/10.1021/acs.energyfuels.4c03095 and https://doi.org/10.62762/JGEE.2025.365363. The literature review should not be limited solely to these references.

3. The quality of the figures is relatively poor, with some data points being unclear. Please standardize and adjust the figures accordingly.

4. The figures should be embedded within the main text of the manuscript. Otherwise, it is challenging for reviewers to cross-reference them during evaluation.

5. Please do not use comment/annotation mode. Instead, incorporate the revisions directly into the text, highlight the changes in red, and upload this version solely as a Supplementary Material file.

6. Finally, please standardize the formatting of the manuscript, particularly the references. The current format does not conform to the journal’s publication requirements.

---

## [Editor Report · Decision Letter 2]

3 Mar 2026

PONE-D-25-39619R2Using fractal model and factor analysis for FACA modeling and its application in deep mineral predictionPLOS One

Dear Dr.  Qin,

Thank you for submitting your manuscript to PLOS ONE. After careful consideration, we feel that it has merit but does not fully meet PLOS ONE’s publication criteria as it currently stands. Therefore, we invite you to submit a revised version of the manuscript that addresses the points raised during the review process.

We look forward to receiving your revised manuscript.

Kind regards,

Hu Li

Academic Editor

PLOS One

Journal Requirements:

Additional Editor Comments:

Dear author, although the reviewers have confirmed the acceptance of this paper, in its current form, the manuscript still requires some improvements before it can proceed to publication. These improvements are mainly reflected in the following aspects:

1. The quality of the figures is too poor, and their formatting is not standardized. I recommend revising them one by one.

2. Please insert the figures and tables into the main text and ensure that each corresponds to the content accordingly.

3. The abstract is too lengthy, particularly the methods and results sections, which can be condensed.

4. The introduction does not sufficiently reflect the current state of fractal model and factor analysis for FACA modeling. In particular, the authors should emphasize how this study differs from previous research, thereby further highlighting the novelty of the paper.

5. The ideal manuscript structure is: 1. Introduction, 2. Materials and Methods, 3. Results, 4. Discussion, followed by Conclusions. All sections and subsections must be numbered properly, e.g., 1. Introduction, 2. ...

6. Please pay appropriate attention to some of the papers already published in the journal and further increase the number of references, such as papers on fractal prediction methods (https://doi.org/10.1007/s40948-026-01117-7).

7. Finally, please strictly follow the journal’s formatting guidelines when preparing the manuscript.

---

## [Editor Report · Decision Letter 3]

16 Apr 2026

Using fractal model and factor analysis for FACA modeling and its application in deep mineral prediction

PONE-D-25-39619R3

Dear Dr. Qin,

We’re pleased to inform you that your manuscript has been judged scientifically suitable for publication and will be formally accepted for publication once it meets all outstanding technical requirements.

Kind regards,

Hu Li

Academic Editor

PLOS One

Additional Editor Comments (optional):

The authors have made multiple revisions to the paper, and all reviewers have confirmed its acceptance. Therefore, the paper can be accepted for publication.
---

## [Editor Report · Acceptance letter]

PONE-D-25-39619R3

PLOS One

Dear Dr. Qin,

I'm pleased to inform you that your manuscript has been deemed suitable for publication in PLOS One. Congratulations! Your manuscript is now being handed over to our production team.

Kind regards,

on behalf of

Pro.Dr. Hu Li

Academic Editor

PLOS One